# Test Research on Residual Mechanical Properties of Fiber-Reinforced Concrete Segments after High Temperature

**DOI:** 10.3390/ma17061418

**Published:** 2024-03-20

**Authors:** Gang Zong, Yao Wang, Yong Wang, Zhaoqing Ren

**Affiliations:** 1College of Architecture and Engineering, Yancheng Polytechnic College, Yancheng 224005, China; 2015230424@yctei.edu.cn; 2Jiangsu Key Laboratory of Environmental Impact and Structural Safety in Engineering, China University of Mining and Technology, Xuzhou 221008, China; 2015220315@yctei.edu.cn

**Keywords:** concrete shield tunnel segment, mixed fiber concrete, high temperature, mechanical properties

## Abstract

In order to research the residual mechanical properties of concrete shield tunnel segments after exposure to high temperatures, two types of concrete segments were designed: a self-compacting concrete segment and a mixed fiber (steel fiber and polypropylene fiber) self-compacting concrete segment. The mechanical properties of seven blocks of concrete segments (five segments after high-temperature exposure and two segments at room temperature) were tested to analyze the influence of different loading sizes and fibers on the development of cracks after high temperature, failure mode, crack width, deformation, and so on in the concrete segments. The results showed that the damage model of the segment after exposure to high temperature and the segment at room temperature were crushed in the pressurized zone, but the high temperature had little effect on the concrete in the pressurized area. The size of the preload at high temperatures had little effect on the remaining load capacity, and the effect on the number of cracks was mainly concentrated on the internal arc surface of the segment. After high-temperature exposure, the number of cracks on the sides and inner arc surface of the segment increased, and the development of cracks was concentrated as several major cracks at high temperatures. When fibers were incorporated, the cracks in the segment became obvious, where the cracks at the loading point became denser and the interval distance became smaller.

## 1. Introduction

Although fires only occur occasionally in shield tunnels, they are extremely harmful [1]. Not only can tunnel fires result in significant property damage and human casualties but may also cause severe damage to the tunnel lining structure [2,3,4,5]. Damage to lining structures reduces their carrying capacity and can even cause the tunnel to collapse. Hence, investigating the mechanical properties of lining pipe sheets following exposure to high temperatures is of paramount importance for conducting post-disaster safety assessments of lining structures.

High-strength concrete is often used in shield tunnel structures, which can easily burst at high temperatures, and fibers (such as steel fiber concrete and polypropylene fiber) are used to reduce their burst behavior and improve post-disaster performance. Zhang et al. [6] studied the effects of external pressure load on steel fiber concrete lining structures. Their findings revealed that external pressure loads significantly influence the damage patterns and crack propagation in segments. Moreover, they can restrict the expansion and deformation of the lining structure under high-temperature conditions. It was also observed that the segment’s edge may fracture spontaneously in the absence of external forces. Hua et al. [7] studied the effects of parameters such as polypropylene fiber, restraint, and fire duration on tunnel lining roof fire damage. The results showed that the addition of polypropylene fiber can reduce the degree of peeling of floor concrete but may increase the number of cracks and residual deformation. Novak et al. [8] conducted high-temperature tests on mixed fiber concrete, and their results revealed that, when the temperature is high, the concrete presents a large number of cracks and an increase in porosity and that the presence of fibers no longer affected the tensile strength of the material. Several scholars [9,10,11,12] carried out experimental research on reinforced concrete and hybrid fiber concrete shield tunnel segments under high-temperature conditions. The results revealed that the use of mixed fibers can effectively provide good anti-spalling properties. Ning [13] studied the mechanical properties and water permeability of mixed fiber self-compacting concrete after exposure to high temperatures. Their results revealed that mixed fiber concrete has fire and water permeability properties comparable to single-mixed steel fiber concrete, and that mixed fibers can significantly enhance the bending strength and bending toughness of concrete.

Mixed fiber concrete exhibits superior mechanical properties to single-doped fiber concrete and reinforced concrete. To date, many scholars have focused most of their high-temperature test research on shield tunnel segment fires on reinforced concrete, steel fiber concrete, and polypropylene fiber concrete [14,15,16] and have obtained a wealth of research results. Steel fiber and polypropylene fiber have different characteristics under high fire temperature conditions, which can improve the performance of the shield segments more comprehensively when they are mixed with concrete. However, to date, there has been relatively limited research on the post-disaster properties of mixed fiber self-compacting concrete segments.

Taking the above into account, in this study, mixed fiber (steel fiber and polypropylene fiber) self-compacting concrete segments were prepared, and the bearing capacity of five self-compacting concrete segments after exposure to high temperature and two self-compacting concrete segments at room temperature were tested. The effects of preloading and mixed fibers on the crack development, failure mode, strain, and deformation of self-compacting concrete segments at high temperatures were analyzed.

## 2. Test Methods

### 2.1. Test Sample Introduction

The cubic concrete strength grade of the segment concrete was C50; the concrete mix proportion refers to the design results of self-compacting concrete given in references [17,18], and its matching ratio is shown in Table 1. Steel fiber can improve the strength of concrete, whereas polypropylene fiber forms pores after being exposed to high temperatures, facilitates the flow of water, and can reduce the vapor pressure of concrete. As shown in Figure 1, 30 kg/m^3^ of steel fiber and 2 kg/m^3^ of polypropylene fiber were added to the mixed fiber concrete. The length of the steel fiber was 35 mm and its diameter was 0.7 mm; the length of the polypropylene fiber was 12 mm and its diameter was 0.03 mm; and the melting point was about 170 °C.

In this study, the mechanical properties of seven concrete segments (five segments after exposure to high temperature and two segments at room temperature) were tested. The five concrete segments that were exposed to high temperature were labeled RC-1-PF, RC-2-PF, HFRC-3-PF, HFRC-4-PF, and HFRC-5-PF, and the two concrete segments at room temperature were labeled RC-6 and HFRC-7. RC denotes self-compacting concrete segments, and HFRC denotes self-compacting concrete segments with mixed fibers.

Based on the requirements set out in the literature [19], the reinforcement ratio used in this study was 0.41%. The strength grade of longitudinal reinforcement on the inner and outer curved surfaces and hoop reinforcement was HRB400, and the reinforcement rate and method of the segments after exposure to high temperature were the same as those of the segments at room temperature. The size and reinforcement of the concrete segments are shown in Figure 2. The production process of the experimental concrete segments was the same as that described in references [20,21]. The production process of the test segments is as follows: (1) steel mold design and fabrication; (2) binding of steel cage and production of concrete thermoelectric couple block: first, place the thermocouple wire in PVC pipe according to the measured spacing, and then, after pouring mortar, bond the mortar block to the steel bar and leave the lead; (3) concreting; and (4) standing maintenance (3 days after removal). The segments were cured in the open air at room temperature for more than 600 days. The loading size, maximum temperature experienced, and maximum deformation of the segments after the disaster at high temperatures are shown in Table 2.

### 2.2. Test Device

The test device was formed with a set of reaction frames, a vertical jack, a horizontal jack, a distribution beam, and a set of supports (the left and right ends of the test sample correspond to fixed hinge supports and movable hinge supports, respectively), and the specific dimensions were the same as those described in reference [22], the test device was shown in Figure 3 (China University of Mining and Technology, Xuzhou, China).

At the movable support end, the horizontal displacement of the segments was limited to form a restraining axial force.

### 2.3. Measurement Methods

#### 2.3.1. Displacement Measurement Method

The arrangement position of the segments after exposure to high temperature was the same as that of the room-temperature segment’s displacement sensor. A differential displacement sensor (LVDT) was used to measure the vertical (horizontal) displacement of the segments. The data were collected using an Agilent34980A data collector. As shown in Figure 4, there are 6 measuring points for vertical displacement of segments. Two measuring points, V3 and V4, were symmetrically arranged in the middle span; the measurement points V2 and V5 were arranged at the third equilibrium cross-sectional position; and two vertical displacement measurement points, V1 and V6, were arranged near the support.

#### 2.3.2. Strain Measurement Method

As shown in Figure 5, CT-1~CT-2 are the concrete strain measurement points in the middle span of the outer arc surface of the segment; CB-1~CB-4 are the concrete strain measurement points in the middle span and loading point of the inner arc surface of the segment; CS-1~CS-3 are the concrete strain measurement points along the thickness direction of the side of the segment, in the middle span, and in the load point position, respectively.

#### 2.3.3. Crack Width Measurement Method

As shown in Figure 6, the crack width was measured using a ZBL-F120 crack width observation instrument (Zhengzhou Zhuotai Testing Equipment Co., Ltd, Zhengzhou, China).

### 2.4. Loading Method

The test segments were loaded through load control; the loading pressure of the jack was manually controlled using the oil pump; and the specific load value was read out from the pressure indicator, as shown in Figure 7. At the same time, the segments were preloaded using 100 kN before the test began, ensuring close contact between the support and the segments.

For 0 kN to 400 kN, 50 kN per level was used for loading; for 400 kN to 600 kN, 30 kN per level was used for loading; and for above 600 kN, 20 kN per level was used for loading until the segments were damaged. When the displacement remained constant and each level was held for 5 min, the next level of loading was carried out. The damage criteria refer to the relevant provisions of references [23].

## 3. Test Phenomena

Figure 8, Figure 9, Figure 10, Figure 11, Figure 12, Figure 13 and Figure 14 show the crack maps and failure characteristics of the side and inner curved surfaces of seven self-compacting concrete segments, in which the red line represents the crack of the concrete segment after being exposed to high temperature, and the red area represents the crushing area after exposure to high temperature.

### 3.1. Segments after Exposure to High Temperature

#### 3.1.1. RC-1-PF

Figure 8a–d show crack layout diagrams of the sides and internal arc surface of segment RC-1-PF, and Figure 8e–f show the diagrams of destroyed features of segment RC-1-PF under limit status. The crack width of the segment’s sides increased under high-temperature conditions and did not generate new cracks before being loaded with 200 kN. The cracks at the load point showed a forking pattern; the load point at the internal arc surface showed symmetry through long cracks; the middle span position exhibited cracks at 350 kN; and the length of both end cracks was about 100 mm. At 440 kN, the cracks of the load point on the duct side gradually extended toward the external arc middle span in the upper-layer concrete iron direction, with no new cracks generated from the load point to the saddle, and the cracks’ width showed almost no change. At the same time, most of the external arc surface cracks were crack extensions under high-temperature conditions, so no new cracks were generated. At 550 kN, the cracks near both sides of the load points were mutually connected. The length of the cracks was 500 mm, causing the tensile failure sound of the concrete (see Figure 8f), and the width of the generated cracks at the arch foot position increased to 5 mm under high-temperature conditions. At 620 kN, there was partial crushing of the external arc surface middle span and concrete at the center position of the load point, along with concrete iron tensile failure sound. When the test finished, in the concrete at the arch feet close to the duct side, there was a large area (up to 0.05 m^2^) of peeling off, and the concrete iron was exposed.

#### 3.1.2. RC-2-PF

Figure 9a–d show crack layout diagrams of the sides and internal arc surface of segment RC-2-PF.

Although the pressure sensor of segment RC-2-PF caused a failure and, as indicated on the display instrument, the number increased, this still did not destroy the segment when it was loaded with 900 kN. The load prior to the increase in temperature was 200 kN, and it generated cracks and made contact with the cracks under high-temperature conditions. At 400 kN, the middle span and loading position experienced new cracks, with a crack length of about 800 mm; the length barely changed after the test was finished. At 420 kN, forking started to occur at the right-end load point of the segment’s sides and at 50 mm from the external arc surface, with the cracks extending in the upper-layer concrete iron direction. At the load point of the internal arc surface, new cracks started to appear through long cracks, and there was obvious forking of the cracks. At 560 kN~650 kN, the cracks of the left-end load point of the segment’s sides extended toward the external arc surface.

#### 3.1.3. HFRC-3-PF

Figure 10a–d are crack layout diagrams of the sides and internal arc surface of segment HFRC-3-PF. At 250 kN, new cracks appeared on the segment middle span, with a crack length of about 60 mm. At 400 kN, the cracks at the left-end load point of the duct side experienced forking and ring cracks which developed along the upper-layer concrete iron toward the middle span saddle direction; the right-end load point experienced new cracks; the internal arc surface load point and middle span broke into multiple pieces through long cracks; and obvious left-side cracks exhibited a forking appearance. At 700 kN~760 kN, arc cracks formed at the upper-layer concrete iron position between both load points of the segment. At 780 kN, when the concrete at the middle span left-side external arc experienced partial crushing, the test was stopped. When the test finished, peeling off did not occur because the larger width of the cracks at the segment right-side saddle position and bridge matched the performance of steel fiber (see Figure 10c).

#### 3.1.4. HFRC-4-PF

Figure 11a–d show crack layout diagrams of the sides and internal arc surface of segment HFRC-4-PF. The segment’s side left-end load point experienced new cracks when the load was increased to 350 kN; the segment’s side right-end load point and middle span position experienced new cracks, and these increased on the internal arc surface at 430 kN. At 540 kN, the duct side middle span cracks developed upward, with a crack length exceeding 100 mm, and the right end’s new cracks also experienced an upward extension. At 680 kN, arc cracks forming multiple pieces at the segment’s side left and right load points developed under high temperature, and forking was evident on the cracks on the internal arc surface. At 780 kN, there was partial crushing of the concrete at the external arc surface left load point. When the test finished, the concrete at the right-side saddle middle span position peeled off, with a peeling-off area of 0.03 m^2^.

#### 3.1.5. HFRC-5-PF

Figure 12a–d show crack layout diagrams of the sides and internal arc surface of segment HFRC-5-PF. When loaded with 250 kN~380 kN, the new cracks at the segment’s sides were mainly concentrated at the left load point and middle span position, and the crack width generated under high temperature increased. When loaded with 440 kN~580 kN, unlike HFRC-3-PF and HFRC-4-PF, the arc cracks at the segment’s side left load point developed toward the left side of the external arc surface. Additionally, the cracks at the left load position of the internal arc surface were more concentrated, but there were fewer cracks on the right side. When, with 680 kN, the external arc surface left load area experienced partial crushing, the loading was stopped.

### 3.2. Room-Temperature Segments

#### 3.2.1. RC-6

Figure 13a–d show crack layout diagrams of the sides and internal arc surface of segment RC-6, and Figure 13e shows a diagram of destroyed features of segment RC-6 under limit status. At 100 kN, the cracks on the segment’s sides were first generated at the load point; at 150 kN, the number of cracks at the segment’s side load point increased and cracks formed in the middle span area. At 300 kN~500 kN, cracks on the left and right load points of the segment’s sides extended in the middle span and external arc surface direction, and partial cracks developed on the left side in the external arc surface saddle position. At 600 kN, multi-piece symmetry was observed through long cracks on the internal arc surface at the segment’s side right load point. At 680 kN, a large amount of crushing was observed in the concrete at the external arc surface middle span area, to a greater crushing degree; the peeling-off area of the concrete at the right-side saddle position was 0.054 m^2^ after the test finished, but the concrete iron was not exposed. During loading, the crack development process of the segment was basically consistent with that described in reference [24,25].

#### 3.2.2. HFRC-7

Figure 14a–d show crack layout diagrams of the sides and internal arc surface of segment HFRC-7 under room-temperature conditions, and Figure 14e shows a diagram of destroyed features of segment HFRC-7 under limit status.

At 150 kN, the segment’s side right load point generated cracks. At 200 kN, cracks formed on the segment’s side middle span area, and cracks at the load point extended upward. At 300 kN, forking occurred on the cracks at both side load points, even in the interval distances between the cracks. At 400 kN, long cracks formed at the internal arc surface right load point. At 500 kN~700 kN, the cracks at the segment’s side left load point extended toward the middle span, but the cracks at the right load point extended toward both sides. At 880 kN, the external arc surface left load point experienced partial crushing.

The number of cracks and their width for each test segment were different, the segment crack-related parameters are shown in Table 3.

### 3.3. Contrast Analysis

Contrast analysis of the crack development and failure mode of the five segments after exposure to high temperature (RC-1 to HFRC-5) and two room-temperature segments (RC-6, HFRC-7) was conducted and the following conclusions were drawn:(1)The damage mode of the segments after exposure to high temperature and the segments at room temperature were crushed in the pressurized zone, and the high temperature had little effect on the concrete in the pressurized area, so the bearing capacity after high temperature was reduced by 10% to 20%. For fiber-free segments, incorporating fiber increased the carrying capacity of the room-temperature segments by 28%, indicating that in the case of a low reinforcement ratio (0.44%), mixed fibers improved the mechanical properties of the segments significantly. Meanwhile, compared with the RC-1 and HFRC-3 test samples, mixed fiber increased the ultimate bearing capacity of the segments by 19.3%. This shows that in mixed fiber segments after exposure to high temperature, due to the melting of polypropylene fibers, the addition of fibers reduced the increase in the remaining bearing capacity of the segments.(2)Contrasting the segments HFRC-3 to HFRC-5, it can be seen that the size of the preload at high temperature had little effect on the remaining loading capacity, with losses of 16%, 11%, and 22%, respectively. The impact on the number of cracks was mainly concentrated on the inner arc surface. It should be noted that, compared with the RC-1 and HFRC-3 test samples, the bearing area of the segments bearing-loaded with a positive bending moment was mainly concentrated at the arch foot (joint); the bridging effect of the steel fiber reduced the crushing of the concrete at the arch foot, which shows that the addition of steel fiber can increase the local strength of the weak part of the lining sheet. Meanwhile, the bearing capacity of the segments in the literature [26,27,28] is limited. The bearing capacity of the segments is limited due to pure bending, and the axial force on the opening load and the extreme load of the segments are greatly increased.(3)After exposure to high temperature, the number of cracks on the side and inner arc surface of the segments increased, and the crack development was concentrated in a few major cracks formed at high temperature. Meanwhile, compared with fiber-free duct segments, mixed fiber made the cracks in the segments obvious; the cracks at the loading point were denser; and the gap between cracks was reduced.

## 4. Test Results

### 4.1. Load–Middle Span Displacement Curve

Figure 15a–g show the load–middle span vertical displacement curves. Among them, V-3 and V-4 are vertical deformations in the middle span of both ends of the segment distribution beam.

Comparing Figure 15a–g, it can be seen that, on one hand, the load–middle span displacement curve of each segment is the same. At 500 kN to 600 kN, the slope of the curve of each piece decreases. However, when the segments were damaged, the curve of each was non-linear. On the other hand, compared with the room-temperature segments, the ultimate deformation of the undamaged fiber segments was reduced, while the ultimate deformation of the segments mixed into the fiber was increased. For example, the ultimate deformations of the HFRC-3-PF~HFRC-5-PF segments were 20.3 mm, 18.84 mm, and 22.29 mm, respectively, while the ultimate deformation of the HFRC-7 segment was 20.1 mm.

Table 4 lists the failure mode, ultimate bearing capacity, and corresponding ultimate displacement of the seven segments. It can be seen from the table that the difference in the ultimate deflection of the segments was small and that all of the deflections were between 1/80 (19.1 mm) and 1/60 (25.0 mm) of the length of the inner arc surface. Therefore, the failure criterion given in the literature [29,30] that the displacement of the segment in the whole fire process should be less than the clear span L/30 (54.2 mm) overestimates the bearing capacity of the segment.

### 4.2. Load–Strain Curve

#### 4.2.1. Load–Strain Curves of Concrete on the Sides of the Segments

Figure 16a–g show the load–concrete strain curves on the sides of the segments, where CS1 and CS3 are the concrete strain near the loading point, and CS2 is the concrete strain across the middle span. The specific measurement point location is shown in Figure 5. It can be seen from the diagram that, on one hand, before concrete cracking, the tensile strain of concrete changes little. After concrete cracking, the strain near the loading point of the segments surges after being exposed to high temperatures, and the strain growth rate increases. When the crack width is large, the strain pieces break down, while the room-temperature segment strain first develops in the middle region. Meanwhile, the concrete strain on the side of the segments close to the outer curved surface shifts from being pressurized to being pulled.

On the other hand, the addition of mixed fibers effectively delayed the development of concrete strain before cracking. For example, at 200 kN, the maximum tensile strain values of RC-1-PF and HFRC-4-PF were 3352 × 10^−6^ and 1153 × 10^−6^, respectively.

#### 4.2.2. Load–External and Internal Arc Surface Strain Curves

Figure 17a–g show the load–concrete strain curve for the outer and inner curved surfaces of the segments, where T represents the outer arc and B represents the inner arc surface, the positive value is pulled, and the negative value is pressurized.

As can be seen from the figure, on the one hand, the pressure strain on the outer arc surface of the segments changes linearly with the increase in load, which is consistent with the development trend of the load–deformation curve. Meanwhile, compared with fiber-free segments, the maximum pressure strain of the segments is increased when incorporating fiber. For example, the maximum pressure strain values of RC-1-PF and HFRC-3-PF were 1480 × 10^−6^ and 2145 × 10^−6^, respectively.

On the other hand, when the segments are damaged after being exposed to high temperatures, the maximum pressure strain is greater than that of the room-temperature segments [31,32,33,34]. For example, the peak pressure strain of HFRC-3-PF to HFRC-5-PF and HFRC-7 were 2145 × 10^−6^, 2781 × 10^−6^, 2413 × 10^−6^, and 1430 × 10^−6^, respectively.

#### 4.2.3. Load–Reinforcement Strain Curve

Figure 18a,b show the load–reinforcement strain curves. Among them, SA represents the longitudinal bar in the middle span, SB represents the longitudinal bar at the loading point, and the positive (negative) value is pulled (compressed).

As can be seen from Figure 18a,b, the unblended segments and the mixed fiber segments yield when loaded with 250 kN and 300 kN, respectively, indicating that the addition of fiber reduces the strain on the longitudinal bars, and the fibers can share part of the tensile stress. After the steel bars yield, the load–strain curve shows obvious non-linear changes, and the strain surges.

### 4.3. Load–Crack Width Curve

Figure 19a,b show the corresponding crack widths for different load levels of the segments after being exposed to high temperatures and at room temperature.

Compared with concrete segments at room temperature, below 400 kN, the loading point and crack width in the middle of the segments at different load levels after exposure to high temperatures are the same. When the load exceeds 400 kN, the crack width at the loading point increases significantly; compared with fiber-free segments, the load is shorter. Due to the influence of steel fibers, the crack width is smaller at different loading stages, and, when the load is large, the crack width changes significantly, which shows that the influence of fiber on the crack width decreases.

## 5. Conclusions

In this study, the mechanical properties of five concrete segment blocks exposed to high temperature and two concrete segment blocks that remained at room temperature were tested. The main conclusions drawn are the following:(1)The load–middle span displacement change rules for each segment are the same, and the increase in displacement slows down between 500 kN and 600 kN. Whether the segment had been exposed to room temperature or high temperature, the ultimate deformation of unmixed fiber segments was lower than the limited deformation of those with incorporated mixed fibers; thus, the addition of mixed fibers can increase the extreme deformation of the segment.(2)As for the concrete strain on the side of the segment, before the concrete cracked, the change in the tensile strain of the concrete was relatively small. After the concrete cracked, following exposure to high temperature, the strain surge increased, while the room-temperature segment strain first developed in the middle span region. Meanwhile, the addition of mixed fibers delayed the development of the concrete strain before cracking.(3)For the strain of concrete on the outer and inner curved surfaces of the segment, the pressure strain on the outer curved surface of the segment changed linearly with an increase in load, consistent with the development trend of the load–deformation curve. Meanwhile, compared with the fiberless segments, the maximum pressure strain value of the tube edge incorporating fibers was greater.(4)For the strain of steel bars, the yield strength of steel bars without fiber segments was lower than those with fiber segments, indicating that fiber incorporation bears part of the tensile stress and reduces the strain of longitudinal bars.(5)For concrete segments after exposure to high temperature, when loading did not exceed 400 kN, the crack width at the loading point and middle span position under different load levels was the same; however, after loading more than 400 kN, the crack width at the loading point increased obviously. For mixed fiber concrete segments, when the load was large, the influence of the fiber on the crack width decreased, and the crack width changed obviously.

## Figures and Tables

**Figure 1 materials-17-01418-f001:**
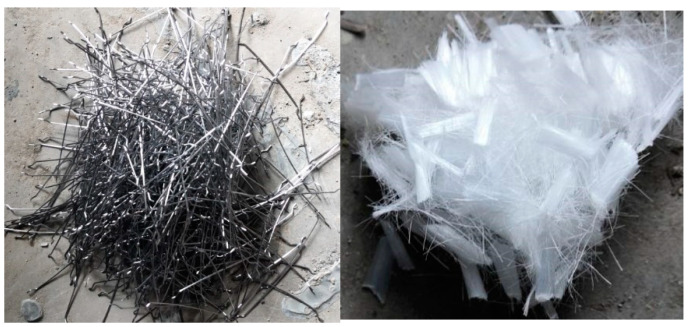
Steel fiber and polypropylene fiber.

**Figure 2 materials-17-01418-f002:**
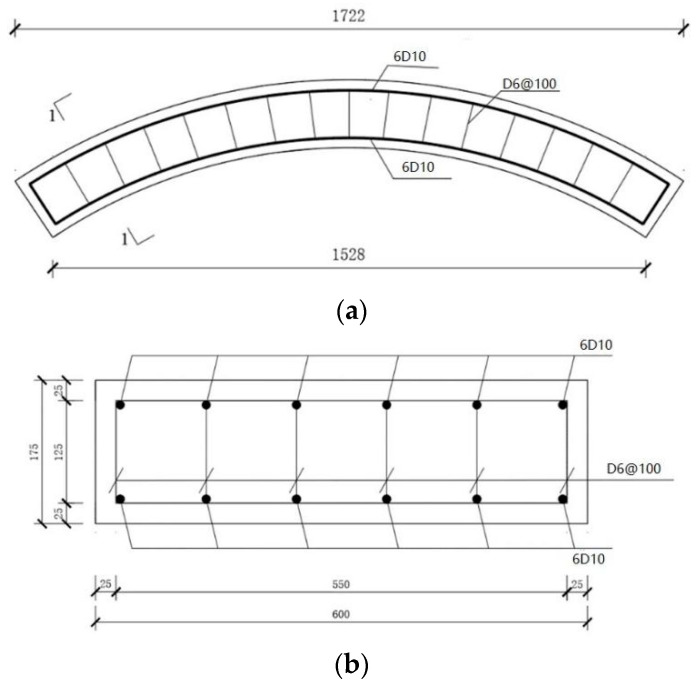
Test segment size and reinforcement diagram (unit: mm). (**a**) Segment size and reinforcement method. (**b**) 1-1 cross-sectional view.

**Figure 3 materials-17-01418-f003:**
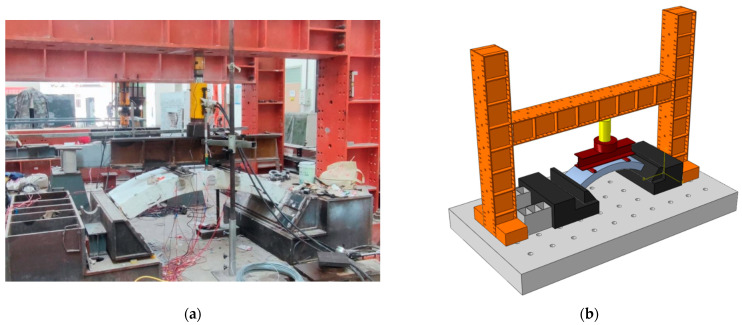
Restraint device and loading device. (**a**) Real view of the test device. (**b**) Three-dimensional diagram of the test device.

**Figure 4 materials-17-01418-f004:**
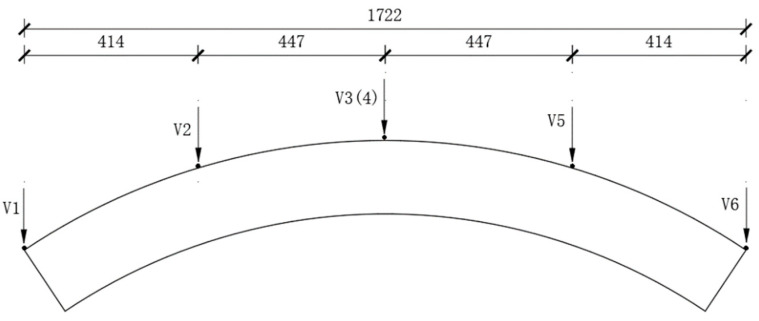
Layout of displacement measurement points (unit: mm).

**Figure 5 materials-17-01418-f005:**
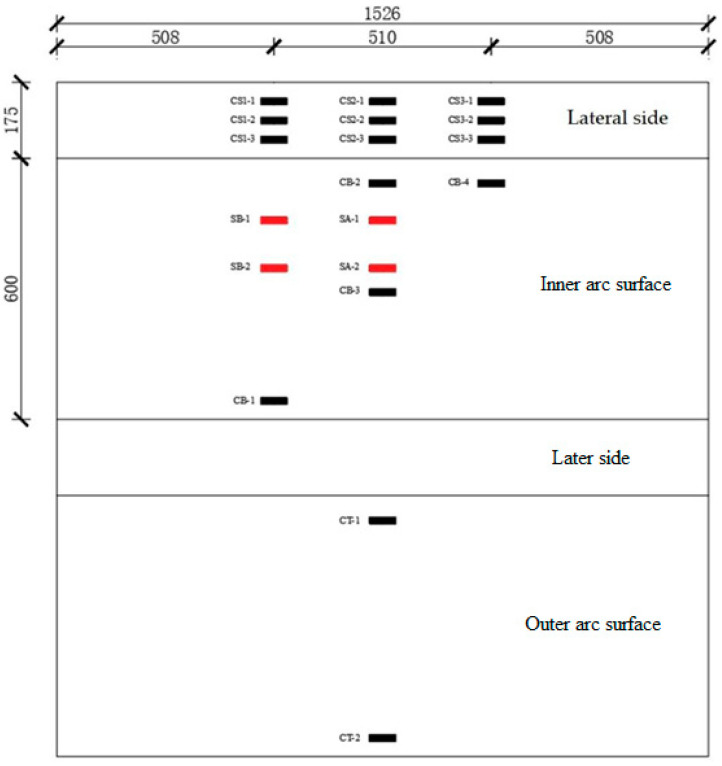
Arrangement of strain pieces (unit: mm).

**Figure 6 materials-17-01418-f006:**
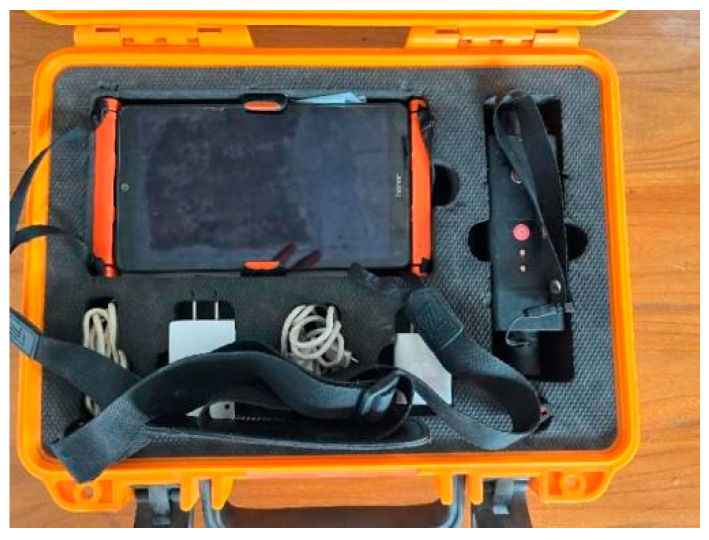
Crack width observation instrument.

**Figure 7 materials-17-01418-f007:**
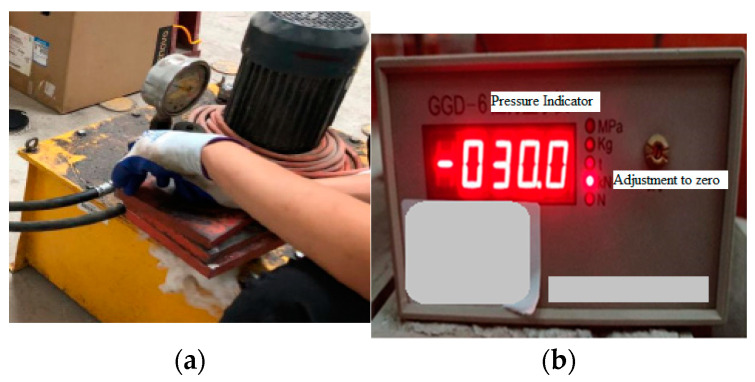
Test loading instrument. (**a**) Hydraulic oil pump. (**b**) Pressure display instrument.

**Figure 8 materials-17-01418-f008:**
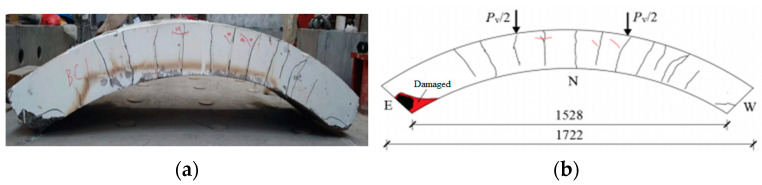
RC-1-PF segment crack distribution map (unit: mm). (**a**) Real view of cracks on the side of the segment. (**b**) Cracks on the north side of the segment. (**c**) Real view of cracks in the curved surface of the segment. (**d**) Cracks in the inner arc of the segment. (**e**) Real view of crushed arch foot. (**f**) Crack on the south side of the segment and failure mode.

**Figure 9 materials-17-01418-f009:**
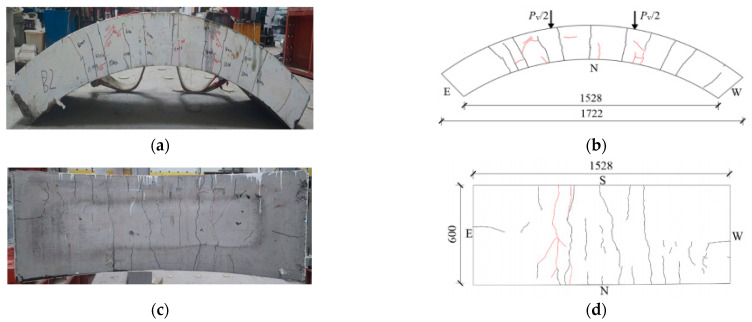
RC-2-PF segment crack distribution map (unit: mm). (**a**) Real view of cracks on the side of the segment. (**b**) Cracks on the north side of the segment. (**c**) Real view of cracks in the curved surface of the segment. (**d**) Cracks in the inner arc of the segment.

**Figure 10 materials-17-01418-f010:**
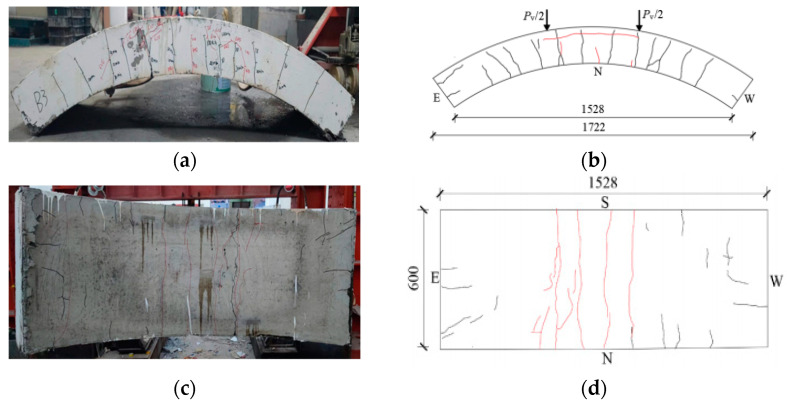
HFRC-3-PF segment crack distribution map (unit: mm). (**a**) Real view of cracks on the side of the segment. (**b**) Cracks on the north side of the segment. (**c**) Real view of cracks in the curved surface of the segment. (**d**) Cracks in the inner arc of the segment.

**Figure 11 materials-17-01418-f011:**
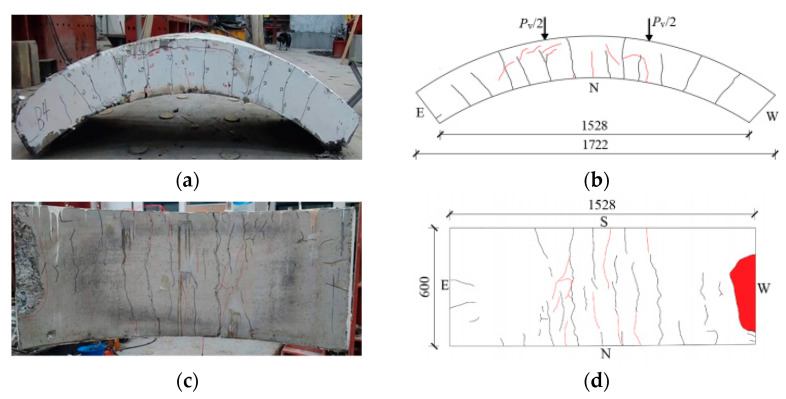
HFRC-4-PF segment crack distribution map (unit: mm). (**a**) Real view of cracks on the side of the segment. (**b**) Cracks on the north side of the segment. (**c**) Real view of cracks in the curved surface of the segment. (**d**) Cracks in the inner arc of the segment.

**Figure 12 materials-17-01418-f012:**
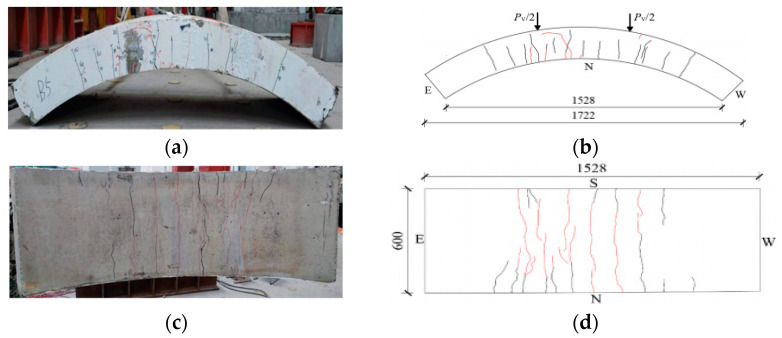
HFRC-5-PF segment crack distribution map (unit: mm). (**a**) Real view of cracks on the side of the segment. (**b**) Cracks on the north side of the segment. (**c**) Real view of cracks in the curved surface of the segment. (**d**) Cracks in the inner arc of the segment.

**Figure 13 materials-17-01418-f013:**
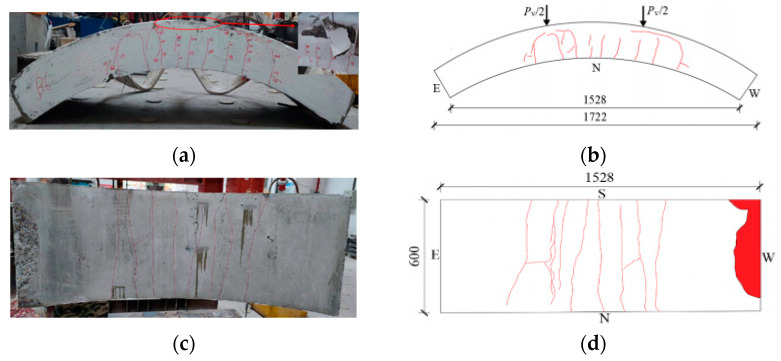
RC-6 segment crack distribution map (unit: mm). (**a**) Real view of cracks on the side of the segment. (**b**) Cracks on the north side of the segment. (**c**) Real view of cracks in the curved surface of the segment. (**d**) Cracks in the inner arc of the segment. (**e**) Crushed compression zone of the segment.

**Figure 14 materials-17-01418-f014:**
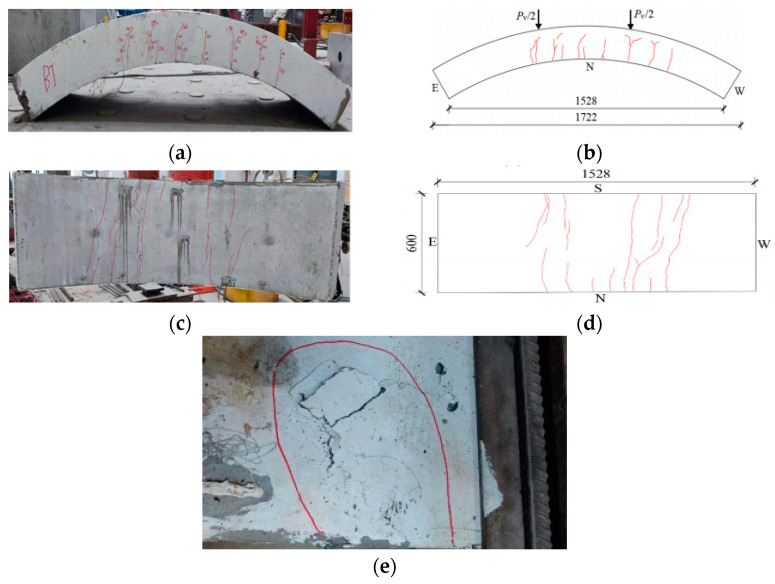
HFRC-7 segment crack distribution map (unit: mm). (**a**) Real view of cracks on the side of the segment. (**b**) Cracks on the north side of the segment. (**c**) Real view of cracks in the curved surface of the segment. (**d**) Cracks in the inner arc of the segment. (**e**) Local crushing at the loading point of the compression zone of the segment.

**Figure 15 materials-17-01418-f015:**
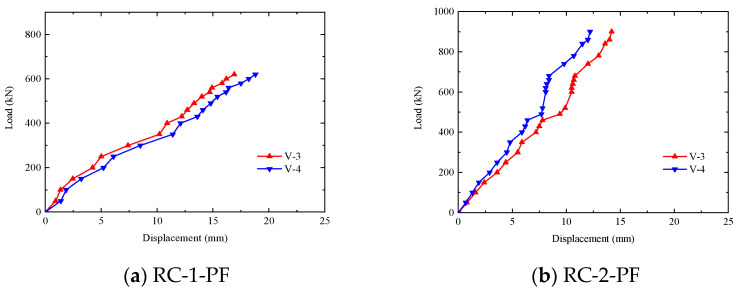
Segments’ load–vertical displacement curves in the middle span.

**Figure 16 materials-17-01418-f016:**
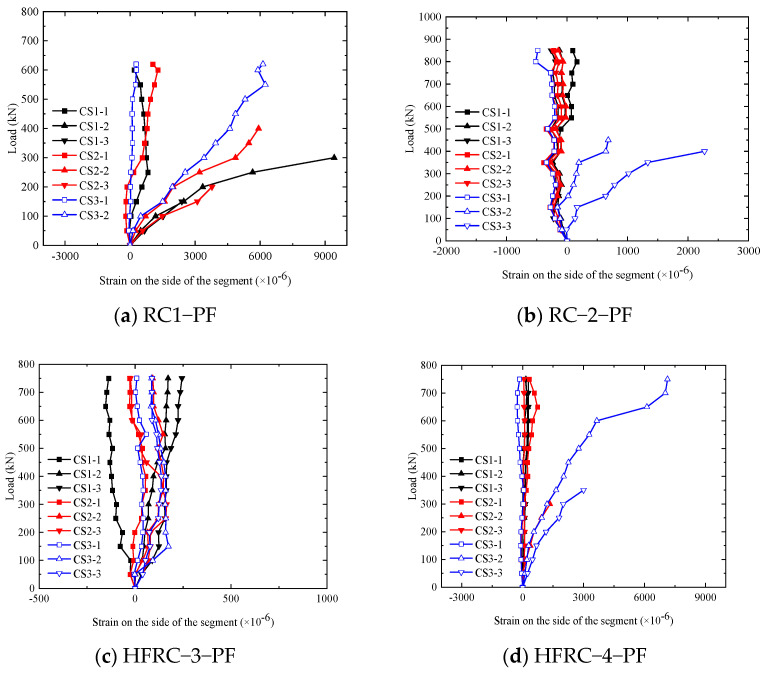
Load–strain curves of concrete on the sides of the segments.

**Figure 17 materials-17-01418-f017:**
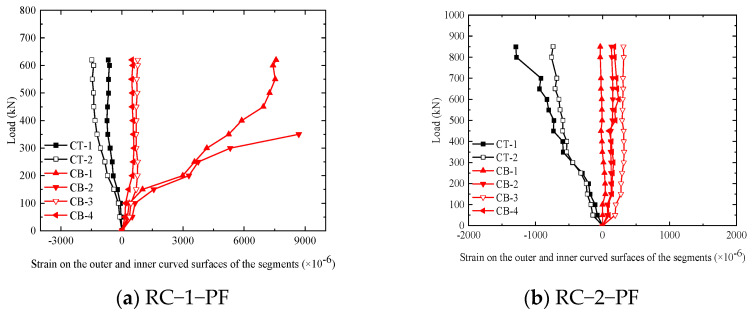
Load–concrete strain curves of the outer and inner curved surfaces of the segments.

**Figure 18 materials-17-01418-f018:**
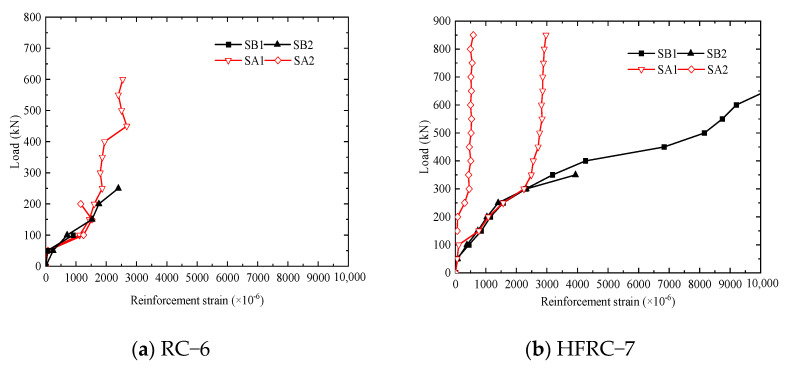
Load–reinforcement strain curves.

**Figure 19 materials-17-01418-f019:**
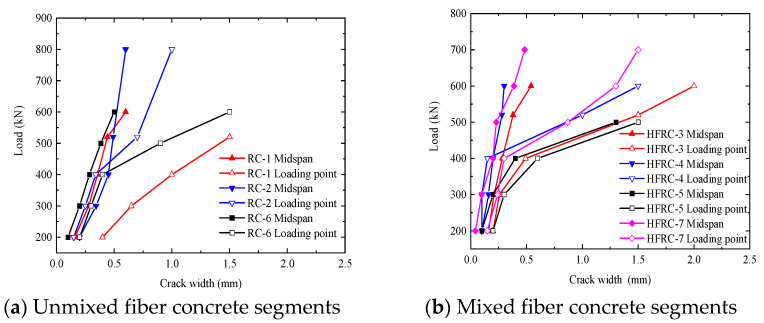
Load–crack width curves of concrete segments.

**Table 1 materials-17-01418-t001:** Mixing ratio of segment concrete (unit: kg/m^3^).

Water	Cement	Sand	Stone	Powdered Coal Ash	Additives
180	400	765	832	160	4.5

**Table 2 materials-17-01418-t002:** Grouping of test segments after exposure to high temperature.

Test Sample Number	Heating Time (min)	Load Size at High Temperature	Concrete (°C)		Steel Bar (°C)		Maximum Deformation at High Temperature (mm)
			Bottom	Top	Lower Layer	Upper Layer	
RC-1-PF	240	PV = 60 kN	471	113	405	127	2.47
RC-2-PF	180	PV = 120 kN	549	96	380	115	4.68
HFRC-3-PF	180	PV = 60 kN	533	94	368	115	2.68
HFRC-4-PF	180	PV = 120 kN	501	102	351	135	5.44
HFRC-5-PF	150	PV = 180 kN	460	101	397	117	5.60
RC-6	room temperature						
HFRC-7	room temperature						
Average value			503	101	380	122	

**Table 3 materials-17-01418-t003:** Segment crack-related parameters (represented by the north side of the segment).

Test Segment Number	Number of Cracks	Average Crack Width (mm)
RC-1-PF	3	1.05
RC-2-PF	5	0.80
HFRC-3-PF	4	1.30
HFRC-4-PF	5	0.90
HFRC-5-PF	4	1.40
RC-6	8	1.00
HFRC-7	7	1.00
Average value	5	1.06

**Table 4 materials-17-01418-t004:** Related parameters and failure characteristics of segments in the limit state.

Test Sample Number	Ultimate Load/kN	Ultimate Deflection/mm	Failure Characteristic
RC-1-PF	620	16.9	Concrete crushing
RC-2-PF	—	—	—
HFRC-3-PF	740	20.3	Concrete crushing
HFRC-4-PF	780	18.84	Concrete crushing
HFRC-5-PF	680	22.29	Concrete crushing
RC-6	680	19.5	Concrete crushing
HFRC-7	880	20.1	Concrete crushing

## Data Availability

Data are contained within the article.

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
