# Peer review of "Test Research on Residual Mechanical Properties of Fiber-Reinforced Concrete Segments after High Temperature"

_materials, 2024, doi:10.3390/ma17061418_

Round 1
Reviewer 1 Report
Comments and Suggestions for Authors
Detailed comments for the authors can be found inside the attached PDF file.

Comments on the Quality of English Language
Minor improvements to English language are needed.
Reviewer 2 Report
Comments and Suggestions for Authors
The document addresses a problem of fundamental importance. However, some essential descriptions are missing. It is not clear, for example, in what way the article is innovative compared to investigations by other scholars. Furthermore, the paper talks about high temperature tests, but it is not clear to what temperature the specimens were brought and for how long. The production process of the experimental pipe segment is referred to articles cited in the bibliography, but it would be useful to include a brief summary of the process in the text. One figure is truncated. Other figures contain non-English text. The general impression is that of a set of data collected hastily, without dedicating the necessary care and in-depth analysis to the presentation of the experimental results.
Line 79
“After high temperature”
Specify how high the temperature was.
Figure 2
There are some symbols to check.
Lines 102-103
“As shown in Figure 4, There are 6 measuring points for vertical displacement of segments.”
The figure shows only 5 measuring points.
Figure 5
There are some symbols and text to check.
Comments on the Quality of English Language
Moderate editing of English language required.
Reviewer 3 Report
Comments and Suggestions for Authors
1. The approach presented in the article is reasonable although was already described in the literature but the investigation in that specific loading condition is valuable. However the article needs several improvement.
2. Could you specify what kind of steel fibres were used in the experiment (size) and what is the propylene fibres melting point?
3. Line 72. There is something wrong with the notation. [Table1.30kg/m3]
4. Line 82-84 Isn’t the predicate missing? [and method of the segments] please read carefully
5. Line 84-86 [The size and reinforcement of the segments and the segments at room temperature after high temperature are shown in Figure 2.] please read carefully
6. Line 86 why “pipe segment”?
7. Figure 4. shows just five measuring points (V6 is missing)
8. Line 112-114- S-1 to S-4 are strain measurement points on the middle of the span and load points. S-1 to S-4 are strain measurement points on the main ribs of the middle span - please read carefully
9. I assume that Fig 8b is the north side of the segment(see Fig 8f).
10. I am not sure if Fig 10e, f are diagrams and other photos as well.
General remark, I assume that your work would more valuable if your literature review go outside Chinese researchers only. This topic was investigated by many scientist in Europe or the States. Could you please confront with their observations?
Comments on the Quality of English Language
I think that your comments on the results, chapter 3.1 and 3.2 need some more improvement. Did you consider table arrangement for example.
Please check English (for example line 12 shouldn’t it be plural)
Reviewer 4 Report
Comments and Suggestions for Authors
In the Reviewer opinion the article “Test Research on Residual Mechanical Properties of Mixed Fiber Concrete Segments after High Temperature Flexural behavior of a new precast insulation mortar sandwich panel” is good.
In order to research the remaining mechanical properties of concrete shield tunnel segments after high temperature, two types of concrete segments, one is a self-compacting concrete segment, and the other is a mixed fiber (steel fiber and polypropylene fiber) self-compacting concrete segment was designed. The mechanical properties of 7 blocks of concrete segments (5 segments after high temperature and 2 segments at room temperature) were tested to analyze the influence of different loading sizes and fibers on the development of cracks in the concrete segment after high temperature, failure mode, crack width, deformation, etc. The results showed that the damage model of the segment after high temperature and the segment at room temperature were crushed in the pressurized zone, but the high temperature had little effect on the concrete in the pressurized area.
Some comments which greatly enhance the understanding of the paper and its value are presented below. Specific issues that require further consideration are:
- The title of the manuscript is matched to its content but it is too long.
- The Introduction generally covers the cases.
- The methodology was clearly presented.
- In the Reviewer’s opinion, the current state of knowledge relating to the manuscript topic has been presented, but the author's contribution and novelty are not enough emphasized.
- Experimental program and results looks interesting and was clearly presented.
- In the Reviewer’s opinion, the bibliography, comprising 21 references, is more less representative.
- An analysis of the manuscript content and the References shows that the manuscript under review constitutes a summary of the Author(s) achievements in the field.
- In the Reviewer’s opinion the manuscript is well written, and it should be published in the journal after minor revision.
Comments on the Quality of English Language
In the Reviewer opinion the article “Test Research on Residual Mechanical Properties of Mixed Fiber Concrete Segments after High Temperature Flexural behavior of a new precast insulation mortar sandwich panel” is good.
In order to research the remaining mechanical properties of concrete shield tunnel segments after high temperature, two types of concrete segments, one is a self-compacting concrete segment, and the other is a mixed fiber (steel fiber and polypropylene fiber) self-compacting concrete segment was designed. The mechanical properties of 7 blocks of concrete segments (5 segments after high temperature and 2 segments at room temperature) were tested to analyze the influence of different loading sizes and fibers on the development of cracks in the concrete segment after high temperature, failure mode, crack width, deformation, etc. The results showed that the damage model of the segment after high temperature and the segment at room temperature were crushed in the pressurized zone, but the high temperature had little effect on the concrete in the pressurized area.
Some comments which greatly enhance the understanding of the paper and its value are presented below. Specific issues that require further consideration are:
- The title of the manuscript is matched to its content but it is too long.
- The Introduction generally covers the cases.
- The methodology was clearly presented.
- In the Reviewer’s opinion, the current state of knowledge relating to the manuscript topic has been presented, but the author's contribution and novelty are not enough emphasized.
- Experimental program and results looks interesting and was clearly presented.
- In the Reviewer’s opinion, the bibliography, comprising 21 references, is more less representative.
- An analysis of the manuscript content and the References shows that the manuscript under review constitutes a summary of the Author(s) achievements in the field.
- In the Reviewer’s opinion the manuscript is well written, and it should be published in the journal after minor revision.
Round 2
Reviewer 2 Report
Comments and Suggestions for Authors
The Authors responded to all comments except those on the production process. The reviewer remains of the opinion that it would be better to include a brief summary of the process in the text. Alternatively, the Authors could describe the production process in the appendix or in the supplementary material.
Comments on the Quality of English Language
Minor editing of English language required.
Reviewer 3 Report
Comments and Suggestions for Authors
Dear authors, well done improvement. Just hope for small editing activity (do not separate drawing and the caption at different pages, once you are writing methods from capital letter once not (in the chapter titles) and please unify caption letter – fig. 4, anyway fig. 4 needs some more attention (what is d) or c)?)).
